# Negative Effects of Chronic High Intake of Fructose on Lung Diseases

**DOI:** 10.3390/nu14194089

**Published:** 2022-10-01

**Authors:** Adrián Hernández-Díazcouder, Javier González-Ramírez, Fausto Sanchez, José J. Leija-Martínez, Gustavo Martínez-Coronilla, Luis M. Amezcua-Guerra, Fausto Sánchez-Muñoz

**Affiliations:** 1Department of Immunology, Instituto Nacional de Cardiología Ignacio Chávez, Mexico City 14080, Mexico; 2Cellular Biology Laboratory, Faculty of Nursing, Universidad Autónoma de Baja California Campus Mexicali, Mexicali 21100, Mexico; 3Department of Agricultural and Animal Production, Universidad Autónoma Metropolitana Xochimilco, Mexico City 04960, Mexico; 4Master and Doctorate Program in Medical, Dental, and Health Sciences, Faculty of Medicine, Universidad Nacional Autónoma de México Campus Ciudad Universitaria, Mexico City 04510, Mexico; 5Research Laboratory of Pharmacology, Hospital Infantil de Mexico Federico Gómez, Mexico City 06720, Mexico; 6Histology Laboratory, Faculty of Medicine, Universidad Autónoma de Baja California Campus Mexicali, Mexicali 21100, Mexico

**Keywords:** fructose, sugar-sweetened beverages, lung diseases, asthma, chronic obstructive pulmonary disease

## Abstract

In the modern diet, excessive fructose intake (>50 g/day) had been driven by the increase, in recent decades, of the consumption of sugar-sweetened beverages. This phenomenon has dramatically increased within the Caribbean and Latin American regions. Epidemiological studies show that chronic high intake of fructose related to sugar-sweetened beverages increases the risk of developing several non-communicable diseases, such as chronic obstructive pulmonary disease and asthma, and may also contribute to the exacerbation of lung diseases, such as COVID-19. Evidence supports several mechanisms—such as dysregulation of the renin–angiotensin system, increased uric acid production, induction of aldose reductase activity, production of advanced glycation end-products, and activation of the mTORC1 pathway—that can be implicated in lung damage. This review addresses how these pathophysiologic and molecular mechanisms may explain the lung damage resulting from high intake of fructose.

## 1. Introduction

Fructose is a monosaccharide present in fruits, vegetables, honey, and as a constituent of the disaccharide, sucrose. In modern diets, however, the main source of fructose monosaccharide is high-fructose corn syrup (HFCS), which is used as a sweetener in a variety of foods, mostly in sugar-sweetened beverages (SSBs). SSBs are any non-alcoholic water-based beverages with added sugar, including sodas, fruit drinks, sports/energy drinks, pre-sweetened iced tea, and artificially sweetened homemade beverages. Since the 1970s, fructose intake has increased to 7.5% of total energy intake (equivalent to 37.5 g of fructose), due to the substitution of sucrose for HFCS in SSBs [1]. Epidemiological studies have reported that SSBs consumption is high in several countries globally, and often in large amounts, contributing to the overall energy density of diets [2]. A survey of 187 countries found that SSBs consumption in adults is higher in middle-income countries than in high- or low-income countries [3]. In high-income countries such as Australia, the United Kingdom, the United States, and Canada, the consumption of SSBs was 0.55, 0.50, 0.66, and 0.51 servings/day, respectively [3]. In the middle-income countries located in the Caribbean and Latin America (LATAM), the highest consumption of SSBs was observed worldwide, with an average consumption of 1.93 and 1.61 servings/day, respectively, compared to 0.58 servings/day globally [4]. In particular, Mexico presented an average SSBs consumption of 1.21 servings/day [3], which represented 17.5% (325 kcal; equivalent to 81.25 g of sugar) and 19.0% (382 kcal; equivalent to 95.5 g of sugar) of total daily energy intake per capita in children aged 1 to 19 years and adults aged ≥20 years, respectively [5]. This increase in fructose consumption through SSBs has caused the rise of non-communicable diseases (NCDs), including obesity, non-alcoholic fatty liver disease, cardiovascular diseases (CVD), type 2 diabetes mellitus (T2DM), some cancer types, and lung diseases [6]. The Caribbean and LATAM have the highest mortality from NCDs related to SSB consumption worldwide, with around 159 deaths per million adults (compared to 48 deaths per million adults globally) [7]. However, the relationship between chronic high intake of fructose and lung damage in non-communicable diseases, such as lung diseases, is not entirely clear.

In this review, we will address epidemiological studies of the effect, on lung disease, of chronic high intake of fructose from SSBs, and the proposed underlying molecular mechanisms—such as the renin-angiotensin system (RAS), uric acid (UA) production, induction of aldose reductase (AR) activity, the induction of the production of advanced glycation end-products (AGEs), and the activation of the mechanistic target of rapamycin complex 1 (mTORC1) pathway—by which fructose could be causing lung damage.

## 2. The Effect of the Amount of Fructose Intake from SSBs on Non-Communicable Diseases

Fructose is present in nearly everyone’s diet. Epidemiological studies have related high/moderate/low and acute/chronic intake of fructose from SSBs to NCDs. For example, subjects with intake of SSBs in different doses (moderate: 10%, 17.5%; and high: 25% total energy from fructose) for 2 weeks increased their risk of CVD, and developed dose-dependency through rapidly elevated lipid/lipoprotein and uric acid levels [8]. The intake of SSBs (75 g/day of fructose) for 12 weeks, in subjects with abdominal obesity, elevated their liver-fat content and hepatic de novo lipogenesis, which increased the risk of the development of CVD [9]. Healthy, lean subjects, whose intake of fructose from SSBs was 80 g/day for 6 weeks, showed a two-fold increase in hepatic fatty acid synthesis, compared with those who consumed equal glucose concentrations in SSBs [10]. Moreover, subjects who increased their SSBs intake (≤1/week to ≥1/day) for 4 years showed an increase in body weight, along with an increase in the risk of developing diabetes, compared to those who decreased their SSBs intake (≥1/day to ≤1/week) [11]. Similar results were found in subjects with intake of SSBs ≥1/day for 6 years, who presented a high risk of diabetes development, compared with intake of ≤1/month of SSBs [12]. In addition, low and moderate fructose concentration has been observed to induce harmful effects. The intake of SSBs (>50 g/day) in both healthy and type 2 diabetic subjects has been associated with higher hepatic insulin resistance [13]. Subjects with low SSBs intake (40 g/day) for 3 weeks presented an increase in low-density lipoprotein (LDL) particle size and distribution, waist-to-hip ratio, fasting glucose, and inflammatory markers [14]. Likewise, subjects with T1DM, with median intake of 34.6 g/day of fructose, were associated with elevation of plasma triglycerides levels and CVD risk factors [15]. While the evidence indicates that fructose intake has a deleterious effect, there is also evidence to suggest that the effects of fructose intake are dependent on the source of the fructose. A meta-analysis of 13 prospective studies showed no detrimental association between metabolic syndrome (MeS) and fructose intake through yogurt, fruit, 100% fruit juice, or mixed fruit juice [16], and also presented a protective association with incident MeS [16]. Another meta-analysis, of 118 trials, found that fructose intake through fruit improved HbA1c without adversely affecting fasting blood glucose and insulin, compared to SSBs intake [17]. Moreover, it was shown that the ingestion of honey-containing fructose had beneficial effects in T2DM patients, promoting a decrease in body weight, total cholesterol, LDL, and triglycerides, and an increase in high-density lipoprotein (HDL) levels [18]. The evidence suggests that the detrimental effects of high intake of fructose are dependent on the source of the fructose. Thus, fructose intake through fructose-rich industrialized food, such as SSBs, can induce metabolic alterations that can lead to various NCDs.

## 3. Pulmonary Disease in LATAM

Nutrition-related NCDs include chronic obstructive pulmonary disease (COPD) and asthma. Current studies have found associations between SSBs intake and pulmonary diseases [19,20,21]. COPD and asthma are common chronic airway diseases that substantially contribute to morbidity and mortality [22]. COPD affects 210 million people worldwide and is one of the three most common causes of death [23]. LATAM cities presented a COPD prevalence of 19.7% in Montevideo, 16.9% in Santiago, 15.8% in Sao Paulo, 12.1% in Caracas, 8.5% in Bogota, and 7.8% in Mexico City. [24]. LATAM is the geographical area that leads the cause of death from COPD, showing around 23% of premature deaths in the age group of 30 to 69 years in 2010. Specifically, Mexico had a mortality rate of 23.1 per 100,000 inhabitants in 2019 [25]. On the other hand, asthma affected 262 million people worldwide in 2019, and caused 461,000 deaths [26]. In the United States, children and adults had an asthma prevalence of 9.3% and 8%, respectively. In LATAM, the International Study of Asthma and Allergies in Childhood reported that the prevalence of asthma in 599,852,642 children aged 13 to 14 years was 18% [27]. Despite asthma prevalence being higher in high-income countries, most asthma-related mortality occurs in low-to-middle income countries such as LATAM, which presents an asthma mortality of 26.3 per 100,000 deaths [27]. Given that LATAM is a great consumer of fructose through SSBs, the high prevalence of lung diseases (COPD and asthma) in the Caribbean and LATAM may be related to high intake of fructose; however, little is known about this relationship. 

## 4. The Effect of Fructose Intake from SSBs on Lung Diseases 

Current studies have shown that fructose intake from SSBs is associated with alterations in lung function [20,21,28,29,30,31]. A cohort of 1068 healthy mother–infant pairs consuming a mean of 32.5 g/day and 27.8 g/day, respectively, of fructose from SSBs for three months, showed that low intake of fructose from SSBs during pregnancy was associated with increased odds of current asthma in middle childhood (odds ratio [OR], 1.70; 95% confidence intervals [CI]: 1.08–2.67) [20], and in early childhood, low SSB intake for one month was also associated with current asthma in middle childhood, in models adjusted for maternal SSBs (OR, 1.79; 95% CI: 1.07–2.97) [20]. In the Framingham Offspring cohort, it was shown that moderate (2–4 times/week) and high (>5–7 times/week) SSBs intake resulted in a 91% and 61% risk, respectively, of developing asthma, compared with low SSBs intake (1–2 times/month). Moreover, moderate apple-juice and fruit-drink consumers had a 61% and 58% increased risk, respectively, of developing asthma [28]. Another study, in US children aged 2 to 9 years, showed that high SSBs intake (≥5 times/week) presented five times greater odds of asthma development (*p* = 0.012) compared to those with low SSBs intake (≤1 time/month) [29]. Additionally, children with high intake of apple-juice were more than twice as likely to have asthma (OR = 2.43, *p* = 0.035) [29]. A study including 16,907 participants aged 16 years and older showed that participants with moderate (0.1–0.5 L/day) and high (≥0.5 L/day) SSBs intake presented a high prevalence of asthma (14.7% and 14%, respectively) compared to 11.9% of non-SSBs-intaking participants, while those with high SSBs intake showed a high prevalence of COPD (6.0%), compared to 4.2% of non-SSBs-intaking participants [30]. Likewise, in adults aged 20 to 55 years, the prevalence of chronic bronchitis was higher in those with SSBs intake ≥ 5 times/week (8.0%), compared to those with intake  ≤ 1–3 times/month (4.9%) [31]. Conversely, it was observed that fructose intake had the opposite effect if the fructose source was fruit. A study of women age 48–83 years showed that women who had consumed ≥2.5 servings/day of fruit for 10 years had a 37% lower risk of COPD (95% CI: 25–48%; *p* < 0.0001) than those who consumed ≤0.8 servings/day [32]. Another study, of men age 45–79 years, observed that current and ex-smokers who consumed ≥5 servings/day of fruit had a 30% lower risk of COPD (95% CI: 15% to 38%) than those who consumed <2 servings/day [33]. Therefore, the evidence suggests that fructose intake could be involved in the development and worsening of lung diseases.

Studies in animal models have demonstrated lung injury by high intake of fructose. An experimental mouse model showed that an intake of 20% weight/volume (*w*/*v*) fructose in drinking water for 12 weeks altered respiratory mechanics, increasing airway resistance and decreasing lung tissue elastance [21]. Also, the fructose promoted the destruction and remodeling of the lung parenchyma, by increasing the number of mononuclear cells, and decreasing the content of elastic fibers, as well as increasing the production of interleukin (IL)-10, IL-6, and IL-1β, and decreasing the concentration of IL-1Ra in plasma, resulting in the development of pulmonary emphysema [21]. Another study showed that an intake of 10% (*w*/*v*) fructose for 16 weeks increased neutrophil infiltration in the lung and pulmonary fibrosis, due to the presence of connective tissue in the bronchial tree, while the stroma increased the production of collagen fibers [34]. Moreover, an intake of 20% (*w*/*v*) fructose in drinking water for 12 weeks, in mice exposed to cigarette smoke, induced a further increase in alveolar enlargement, along with the recruitment of macrophages and mononuclear cells in the lung tissue, compared to mice exposed only to cigarette smoke [35]. Thus, these studies, in both human and experimental animals, suggest that chronic high intake of fructose is a risk factor for developing lung diseases and their exacerbations. However, the molecular mechanisms by which chronic high intake of fructose through SSBs induces lung inflammatory damage are beginning to be understood. Thus, we will review some of the plausible pathways affected by fructose related to the development of lung damage.

## 5. Uric Acid on Lung Function and Chronic High Intake of Fructose

UA is another plausible lung injury mechanism given its important role in the regulation of lung function. UA is a terminal product of purine metabolism with antioxidant activity, due to its ability to inhibit lipid peroxidation and scavenging reactive oxygen species (ROS) [36]. UA is an important extracellular antioxidant, found in both plasma and lungs [37]. Fructose metabolism induces increased UA levels because it is a byproduct of fructose when it is phosphorylated by ketohexokinase (KHK). Since KHK does not have a negative feedback mechanism, all fructose is phosphorylated until ATP depletion [38]. 

A study of 19 healthy individuals with a mean age of 24 years, and a mean body mass index (BMI) of 24.0 kg/m^2^, showed that intake of SSBs with a high concentration of fructose (75 g) increased plasma UA levels (from 296 to 330 μmol/L) at 15 min, reaching the highest UA levels (361 μmol/L) at 75 min [39]. A study of 16 overweight or obese subjects (BMI 25–35 kg/m^2^), aged 40–70 years, showed that intake of SSBs with high fructose concentration (125 g) for 10 weeks led to an increase of 27 μmol/L of serum UA levels [40]; it was also found that, under 24 h fasting conditions, serum UA levels remained elevated [40]. A randomized controlled trial in 41 healthy participants, with a mean age of 21 years and a mean BMI of 23 kg/m^2^, showed that the intake of SSBs with low fructose concentration (26.7 g) increased plasma UA levels to 22 μmol/L (95% CI, 16–29) and 23 μmol/L (95% CI: 14–33) at 30 and 60 min, respectively, after the intake of SSBs [41]. In addition, a pilot study in patients with chronic kidney disease, type 2 diabetes (DT2) patients, and healthy subjects showed an increase in plasma UA levels from 530 μmol/L to 619 ± 93 μmol/L, from 339 μmol/L to 411 ± 69 μmol/L, and from 317 μmol/L to 350 ± 89 μmol/L at 30 min, respectively, after drinking beverages with a low concentration of fructose (35 g) [42]. Thus, fructose intake (low and high doses) can induce UA production both in healthy subjects and in diseased persons. Although there are no studies that explain the relationship between fructose-produced uric acid and lung injury, existing reports have shown that both low and high levels of UA are associated with impaired lung function, e.g., endothelin-1-stimulated bronchoconstriction [43,44], and oxygen desaturation [45]; low and high levels of UA are also positively associated with markers of systemic inflammation, such as C-reactive protein and IL-6 [46]. Patients with acute lung injury (ALI) showed elevated UA levels in bronchoalveolar lavage fluid, compared to patients without ALI [47]; in addition, UA levels were positively correlated with the mean pulmonary leakage index (*r* = 0.64; *p* < 0.01) [47]. In a study of 86 patients with idiopathic pulmonary artery hypertension (PAH), with a mean age of 35.9 years, the patients with idiopathic PAH were found to have elevated serum UA levels (405 ± 130 μmol/L), and UA was positively correlated with mean pulmonary arterial pressure (*r* = 0.387; *p* < 0.01) [48]. Moreover, a retrospective study of 217 patients with asthma exacerbation found elevated serum UA levels (mild exacerbation: 200.10 ± 44.71 μmol/L; moderate exacerbation: 265.44 ± 62.78 μmol/L; severe exacerbation: 341.54 ± 86.27 μmol/L), and UA was negatively correlated with lung function estimated by peak expiratory flow (*r* = −0.507, *p* < 0.001) [49]. In 314 patients with COPD, high serum UA levels (≥410.41 μmol/L) were related to more severe airflow obstruction, comorbidities, severe dyspnea, and impaired oxygenation, compared to patients with low serum UA levels (<410.41 μmol/L) [50]. The study reported that high serum levels of UA (≥410.41 μmol/L) increased the mortality risk both at 30 days and at 1 year [50]. In patients with severe pulmonary hypertension, high plasma UA levels were found (463.94 μmol/L, IQR 5.80–9.20), and UA levels were positively correlated with pulmonary artery systolic pressure (*r* = 0.51, *p* < 0.001) and the functional class (*r* = 0.49, *p* < 0.001). A case-control study showed that COPD patients showed increasing levels of UA according to COPD severity; the mean UA in mild, moderate, and severe COPD was 4.4, 5.7, and 6.3, respectively [51]. Meanwhile, another study, of 109 stable COPD patients, showed that UA levels were positively associated with inflammatory markers, such as leukocytes (OR: 1.30, 95%; CI: 1.10–1.54), fibrinogen (OR: 1.82, 95%; CI: 1.08–3.07), and IL-1β (OR: 5.10, 95%; CI: 2.15–12.09) [52]. Conversely, there are studies that have shown UA protective effects in female lung injury: for example, in elderly females with the SLC2A9/GLUT9 rs11722228 T/T genotype, it was shown that higher blood UA levels were associated with higher lung function [53]. The administration of 1000 mg of UA (physiological concentration), in patients with T1DM and patients who were smokers, improved endothelial function by increasing vasodilator responses to acetylcholine [54] 

Finally, in animals with bleomycin-induced lung injury, UA was found to exacerbate lung injury, by increasing IL-1β production through nucleotide oligomerization domain (NOD)-like receptor protein 3 (NLRP3) inflammasome activation, resulting in lung inflammation and fibrosis [55]. In a PAH model, increased UA levels in the lungs, following administration of 2% oxonic acid, induced an exacerbation of neointimal occlusive lesions in small pulmonary arteries, resulting in worsening of the PAH [56]. In asthma, aspirin treatment aggravated eosinophilic inflammation in the lung, along with conversion of inflammation from the Th17-type to Th2-type through the adenosine-UA pathway, suggesting that UA is a key mediator in pulmonary inflammation [57]. In the mechanical ventilation-induced lung injury mice model, treatment with uricase (which degrades UA) and allopurinol (xanthine–oxidase inhibitor) did not modify the inflammatory pattern found in lung injury, despite attenuating alveolar-capillary permeability and lung injury by decreasing UA levels [47]. Another study found that febuxostat (xanthine–oxidase inhibitor) reduced pulmonary inflammation by decreasing the neutrophil response [58]. At cellular level, UA treatment induced the expression of intercellular adhesion molecule-1 (ICAM-1) and IL-1β, by activating NLRP3 inflammasome in human umbilical vein endothelial cells (HUVEC) [59]. Likewise, UA treatment in HUVEC cells induced an inflammatory response, by the increase of the expression of NF-κB, MCP-1, ICAM-1, TNF-α, inducible nitric oxide synthase (iNOS), and endothelin-1 (ET-1), resulting in endothelial damage [60]. Conversely, it was found that physiological concentrations of UA have protective effects against injury mechanisms. UA treatment (297 μmol/L) reduced the protein levels of ICAM-1, nitric oxide (NO), and ET-1 in HUVEC cells pre-treated with oxidized (ox)-LDL, resulting in a protective effect against endothelial damages [61]. In addition to the above, studies have found that high UA levels promote lung damage by activating other mechanisms related to inflammation, such as the RAS, aldose reductase (AR), and the receptor for advanced glycation end products (RAGE) [62,63,64,65,66]. Fructose-produced high UA levels, therefore, activate other inflammatory pathways, allowing enhanced inflammatory states, and so high UA concentration promotes lung tissue injury by inducing an inflammatory state. The evidence, then, indicates that high intake of fructose promotes high UA production, which may be involved in the progression of lung injury (Figure 1b). To date, however, the effect of UA induced by high intake of fructose as a mechanism of lung tissue injuries has been poorly studied. 

## 6. The Renin–Angiotensin System on Lung Function and Chronic High Intake of Fructose

One of the molecular mechanisms that plausibly causes lung damage is the activation of the RAS. This system is an important regulator of cardiovascular function. The RAS is constituted by renin, angiotensin (Ang) II, and the angiotensin-converting enzyme (ACE) [67]. In the kidney, juxtaglomerular cells that produce prorenin, the inactive form of renin, are found in the afferent arterioles. Juxtaglomerular cells are activated by decreased blood pressure, β-activation, and sodium load in the distal convoluted tubule. This activation promotes the cleavage of prorenin to renin by the prorenin receptor (PRR) [68]. Renin is secreted into the circulation, where it cleaves to angiotensinogen to form Ang I. Ang II, the active form of angiotensin, is produced through cleavage to Ang I by ACE activity [68]. The biological effect of Ang II includes vasoconstriction, inflammation, apoptosis, vascular infiltration, and fibroproliferation through the interaction of its receptors (Ang-1 receptor [AT1R] and Ang-2 receptor [AT2R]) [69,70]. Conversely, ACE2 acts as a negative regulator of the RAS, by reducing Ang II levels by cleaving to Ang II to generate Ang-(1-7) [71,72]. The effects of Ang-(1-7) oppose most of the actions of Ang II by interaction with the G protein-coupled Mas receptor (MasR) [73].

Abnormal RAS function is involved in volume overload and peripheral vasoconstriction, which increases left ventricular diastolic filling pressures and promotes left ventricular hypertrophy [74,75,76]. The RAS can also be activated in tissues, including the lungs, playing a role in injury response and repair (Figure 1a). For example, RAS activation promotes lung injury, through the development of fibrosis and increased vascular permeability [77,78]. On the other hand, ACE is highly expressed in the lungs, brain, and kidneys [79]. In patients with acute respiratory distress syndrome (ARDS), increased ACE levels in bronchoalveolar fluids are found, reflecting endothelial damage [80]. Additionally, a correlation has been found between ACE activity bound to the pulmonary capillary endothelium, and the severity of lung damage in patients with acute lung injury [81]. The increase in Ang II by ACE activity mediates the inflammatory process through AT1R in lipopolysaccharide (LPS)-induced acute lung injury in rats [82]. In addition, Ang II has been found to increase lung damage by vasoconstriction of pulmonary vessels, increased pulmonary vascular permeability, inflammation, and interstitial fibrosis [77]. Furthermore, ACE2 has been shown to protect against lung injury: for instance, ACE2 knock-out (KO) mice were found to have enhanced vascular permeability, pulmonary edema, neutrophil accumulation, and impaired lung function [77]. In mice with respiratory syncytial virus-induced lung injury, ACE2 reduction and increased Ang II levels in lung tissues are found. Administration of recombinant ACE2 improves lung physiology by reducing alveolar wall inflammation, lymphocyte infiltration, and pulmonary edema [83]. In acid aspiration-induced lung injury in ACE2 KO mice, ACE2 reduced acute lung injury, by decreasing lung elastance and pulmonary edema formation [77]. Furthermore, increasing ACE2 reduced inflammation, through downregulation of the LPS-Toll-like receptor 4 (TLR4) pathway in LPS-induced lung injury in mice [84]. Therefore, the evidence indicates that the ACE–Ang-II–AT1R and ACE2–Ang-(1-7)–MasR axes could play a crucial role in the development of lung injury. 

Although the RAS is known to play a role in lung function, the effect of modulation of the RAS by high intake of fructose is unclear; however, some information suggests a potential association. High intake of fructose is involved in the activation of the RAS in various tissues. For example, recent studies have found that intake of 20% (*w*/*v*) fructose for 12 weeks induces an increase in PRR protein, resulting in elevated renin levels in the kidneys of rats [85]. In mice, the maternal consumption of 20% (*w*/*v*) fructose during pregnancy and lactation promoted an increase in PRR gene expression in the kidneys of offspring [86], and this result may also be found in human kidney cells [85]. Several studies have demonstrated that PRR activity is involved in the inflammatory response. In human endothelial cells, UA exposition induced a rise in the secretion of pro-inflammatory cytokines, by an increase in the activity of PRR [87]. Likewise, PRR induced injury by increasing inflammatory mediators in bronchoalveolar lavage fluid (BALF) in rats, resulting in interstitial edema, hemorrhage, and neutrophil count in the lung tissues [88]. Given that fructose intake is able to induce PRR expression, and that PRR activity is involved in lung tissue injury, the evidence indicates that fructose may induce lung tissue injury by the activation of PRR. Studies are needed, however, to elucidate the effect of fructose on lung tissue injury induced by PRR activity. 

On the other hand, mice that consumed 20% (*w*/*v*) fructose in drinking water during pregnancy transmitted higher serum levels of renin, Ang II, and aldosterone to their offspring, resulting in an increase of blood pressure [89]. Another study found increased expression of the ACE/Ang II/AT1R axis in both the heart and aorta, thereby raising systolic blood pressure, in rats that had consumed 10% (*w*/*v*) fructose for 9 weeks [90]. Moreover, it was found that a diet in which 50% of the total energy intake was in the form of fructose, for 8 weeks, induced right ventricular hypertrophy, due to the imbalance of the ACE/Ang II/AT1R and ACE2/Ang-(1-7)/AT2R axes in murine models [91,92]. A study reported that consumption of 20% (*w*/*v*) fructose for 2 weeks promoted high serum renin and Ang II levels, increased lung ACE expression, and increased renal AT1aR and AT1bR expression [93]. Despite there being no studies indicating the effect of fructose intake on the activation of the RAS in lung injury, some studies have demonstrated that products from fructose metabolism can activate the RAS, and are involved in lung injury. In rats, high intake of fructose (60% of total calories) increased methylglyoxal (MGO)—a metabolite of fructose—production and protein levels of renin, Ang II, and AT1R in the aorta and renal tissue [94]. MGO-induced fructose increases the protein levels of AT1R and Ang II in vascular smooth muscle cells (VSMC) [94]. Likewise, MGO treatment in rat carotid endothelium induced an increase of the Ang II levels, along with an augmented contractile effect [95]. On the other hand, MGO treatment induced an increase in inflammatory response by increasing the airways neutrophil infiltration, along with an elevation of tumor necrosis factor-alpha (TNF-α) and IL-1β levels, in LPS-induced ALI mice model [96]. Likewise, MGO treatment in an acute lung injury induced by an influenza-infection mice model increased mortality and lung area lesion by increasing neutrophils/lymphocytes recruitment [97]. In addition to fructose-activated RAS-induced lung damage, studies have shown that the RAS can activate the receptor for advanced glycation end-products (RAGE), signaling enhanced inflammatory response [98,99]. Evidence suggests that high intake of fructose may induce lung injury through RAS activation (Figure 1a). However, studies are needed to elucidate the effects of fructose intake on the modulation of the RAS in lung tissue, as a driver of lung injury.

## 7. Receptor for Advanced Glycation End-Products in Lung Function and Chronic High Intake of Fructose 

The activation of the RAGE is a potential inflammatory mechanism that participates in the development of lung tissue injuries. The RAGE is a multiligand transmembrane receptor belonging to the immunoglobulin gene superfamily [100]. The RAGE participates in inflammatory responses through the recognition of AGEs and other endogenous ligands [101]. The RAGE exists in two forms: the membrane-bound RAGE (mRAGE) and the soluble RAGE (sRAGE). The sRAGE binds to the RAGE ligands without activating the RAGE-mediated signaling pathway, thus acting as a competitive inhibitor of the RAGE [102]. The RAGE is poorly expressed in most tissues, but highly expressed in the lung, in pulmonary endothelium, bronchial and vascular smooth muscle, alveolar macrophages, leiomyocytes, and the visceral pleural surface [103].

The RAGE plays a role in the pathogenesis of several pulmonary diseases, including cancer and fibrosis [103]. Furthermore, the RAGE has been identified as a marker in acute lung injury [104,105]. In a study of 676 patients with acute lung injury, plasma sRAGE levels were associated with greater severity of lung injury, through positive correlations with higher tidal volume (*r* = 0.12; *p* = 0.002), plateau pressure (*r* = 0.22; *p <* 0.0001), and higher APACHE scores (*r* = 0.17; *p* < 0.0001). High plasma sRAGE levels were associated with increased mortality (OR of death 1.38; 95% CI: 1.13–1.68)) [106]. Another study found that patients with pulmonary hypertension showed elevated plasma levels of sRAGE (patients 1836.9 ± 976.1 pg/mL vs. controls 831.7 ± 361.5 pg/mL) [107], with a positive correlation between plasma sRAGE levels and tricuspid valvular regurgitation pressure gradient (*r* = 0.403, *p* < 0.0001) [107]. A study of individuals from the New York City Fire Department exposed to World Trade Center particulate matter showed that plasma sRAGE levels (≥97 pg/mL) were positively correlated with the development of lung injury (OR = 1.516, *p* = 0.560) [108]. Plasma sRAGE levels (≥97 pg/mL) were found to increase the odds of developing lung injury by 130% [108]. 

On the other hand, in animal models, the RAGE has been shown to play a key role in the development of lung injury [109,110,111]. In a model of *S. aureus*-induced lung injury in mice, the inhibition of high-mobility group box 1 (HMGB1), a RAGE ligand, by antibodies presented a reduction in lung edema, while Rage^−/−^ mice showed a reduced pleuritis [109]. The same study also reported that the anti-HMGB1 treatment showed a decrease in IL-1β and keratinocyte-derived chemokine (KC) levels in BALF, while Rage^−/−^ mice showed low levels of TNF-α, IL-6, and KC [109]. In an acid-injured lung model, recombinant sRAGE and anti-RAGE antibodies treatment attenuated lung injury by improving arterial oxygenation and attenuating alveolar-capillary permeability and inflammation by the decrease of IL-6, TNF-α, KC, macrophage inflammatory protein 2 (MIP-2) and IL-17 in BALF and plasma [111]. Although the RAGE is known to play a role in lung injury, the regulation of the RAGE in lungs by fructose consumption is poorly understood.

A randomized study, including 20 healthy subjects aged 20 to 30 years, showed that an oral fructose load of 40 g decreased serum sRAGE levels (Control: 1327 pg/L vs. Fructose: 1282 pg/L) 3 hours after load fructose, similar to an oral load of 40 g fructose plus 40 g glucose (Control: 1256 pg/L vs. F + G: 1199 pg/L); decrease of the sRAGE levels by fructose consumption may suggest an increase in the mRAGE activity [112]. Thus, fructose intake enhances the deleterious effect of RAGE signaling by decreasing the sRAGE, a competitive inhibitor of the RAGE. It is well known that fructose induces the expression of the RAGE in various tissues. In rats that drank fructose (10% w/v) in their drinking water for 16 weeks, accumulation of AGEs and RAGE mRNA and protein was found to be increased in the cerebral cortex, leading to brain injury by induction of ROS and activation of the factor nuclear factor kappa B (NF-κB) pathway [113]. In a rat model, metabolic syndrome induced by an isocaloric 60% fructose diet for 6 weeks promoted increased plasma AGE levels and increased RAGE protein levels in the kidney and liver, suggesting that the AGE–RAGE axis is involved in the development of fructose-induced metabolic syndrome [114]. Moreover, rats that drank fructose (20% *w*/*v*) in drinking water for 16 weeks showed elevated serum levels of TNF-α, monocyte chemoattractant protein-1 (MCP-1) and AGEs, and an increase in AGEs accumulation and RAGE expression in the gastrocnemius muscle, resulting in impaired insulin signaling [115]. The study also found that treatment with aminoguanidine (a glycation reaction inhibitor) prevented the effects of fructose consumption [115]. At the cellular level, HUVEC cells exposed to fructose-derived AGEs (Fru-AGEs) treatment with anti-AGE DNA aptamer (blocker of Fru-AGE binding to RAGE) prevented ROS generations and vascular cell adhesion molecule 1 (VCAM-1) expression, suggesting that fructose consumption may induce endothelial cell injury in part through activation of the Fru-AGEs–RAGE axis [116]. In L6 myotubes exposed to 15 mM fructose, increase in the accumulation of AGEs and expression of RAGE [115] were seen, suggesting that fructose induces oxidative stress and inflammation in skeletal muscle by activating the Fru-AGEs–RAGE axis [117]. Thus, evidence suggests that high intake of fructose promotes an inflammatory setting in several tissues, by increasing the expression of RAGE. In addition to fructose-produced AGEs as mediators of lung injury, it was found that MGO, a metabolite of fructose, is also a potent former of AGEs, which induced endothelial damage by increasing the expression of inflammatory mediators such as the RAGE, NF-κB, MCP-1, and IL-6 in HUVEC cells [118]. Likewise, it was found that MGO treatment exacerbated T helper 2 cells (Th2)-mediated airway eosinophil infiltration by activation of NF-κB and iNOS in the lung tissue of mice [119]. Given that the activation of the AGE/RAGE axis promotes an inflammatory setting, the evidence suggests that the fructose-activated AGE/RAGE axis is a plausible mechanism for the development and/or worsening of lung tissue injury (Figure 1c). Further research is required, however, for insight into the effect of high intake of fructose on lung tissue injury mediated by the AGE/RAGE axis.

## 8. Aldose Reductase Activity in Lung Function and Chronic High Intake of Fructose

The polyol pathway is a metabolic system in which glucose is reduced to sorbitol and then converted to fructose. The polyol pathway includes two enzymes, AR and sorbitol dehydrogenase [120]. AR (NADPH-dependent oxidoreductase) is the first rate-limiting enzyme for the catalysis of the conversion of glucose to sorbitol [121]. AR also participates in cardiac injury, lung injury, and systemic inflammation [122,123]. For instance, in a cecal puncture-and-ligation mice model transgenic for human AR, AR enhanced the inflammatory response by increasing circulating levels of TNF-α and IL-6 neutrophil infiltration of the lungs by the activation of Rho-kinase in pulmonary endothelial cells [124]. The same study reported that in human lung endothelial cells, AR activity is essential for neutrophil activation, and neutrophil adhesion to endothelial cells induced by stimulation with TNF-α [124]. In a mouse model of asthma, treatment with fidarestat (an AR inhibitor) decreased the infiltration of the number of eosinophils, macrophages, and neutrophils in bronchoalveolar fluid, as well as decreasing eosinophil infiltration in the lungs [125]. Moreover, in human primary small airway epithelial cells, the inhibition of AR activity prevented airway remodeling (epithelial to mesenchymal transition) through the inhibition of phosphatidylinositol 3-kinase (PI3K) and glycogen synthase kinase 3β (GSK3β) phosphorylation induced by transforming growth factor-β1 (TGFβ1) signaling [125]. In a bleomycin-induced pulmonary fibrosis rat model, epalrestat (an AR inhibitor) reduced fibrosis changes in the lung and prevented collagen deposition by decreasing expression and production of TGFβ1, α-smooth muscle actin (α-SMA), and type I collagen [126]. Likewise, the inhibition of AR in primary rat pulmonary fibroblast induced the repression of TGFβ1-induced proliferation of pulmonary fibroblasts, as well as collagen deposition [126]. Thus, studies suggest the important role of AR activity in activating inflammatory response to lung injury.

Studies have demonstrated that fructose can induce the expression of AR in several tissues [127,128,129]. In rats fed a diet containing 64% fructose for eight weeks, AR expression was shown to be increased in the stomach, heart, and lungs [129]. Moreover, in a rat rehydration model, rehydration with 10% (*p*/*w*) fructose was found to promote kidney injury by activating AR [127]. Glucose-fed KHK KO mice showed higher levels of hepatic fructose compared to KHK KO mice in tap water; these results suggest that fructose produced from glucose may induce AR expression in the liver [130], although there is no evidence of lung damage by high-intake-of-fructose-induced AR activity. In the context of high intake of glucose, a study found that HUVEC cells exposed to high glucose concentration (25 mM) induced an increase of ROS production and monocyte adhesion by AR activity, resulting in endothelial damage [131]. This evidence suggests fructose-induced AR activity in the lung (Figure 2a); therefore, studies elucidating the role of fructose-induced AR in lung injury are needed.

## 9. Activation of mTOR Signaling in Lung Function and Chronic High Intake of Fructose

The mammalian target of rapamycin (mTOR) is a ubiquitously expressed multiprotein complex with serine/threonine kinase activity, that plays a central role in cellular growth and metabolism, and is a major downstream effector of the PI3K/Akt pathway [132]. Multiple clinical and experimental studies have suggested that mTOR could be involved in a wide variety of diseases, and clinical trials have been initiated to evaluate the efficacy of mTOR inhibitors in the treatment of several diseases [133].

Specifically, in the lung, it has been observed that mTOR can regulate the biological activities of pulmonary vascular endothelium and alveolar epithelial cells in lung diseases [134,135]. mTOR forms the catalytic subunit of two complexes, mTORC1 and mTORC2 [132]. Evidence indicates that mTORC1 contributes to the pathogenesis of lung injury. A study in ARDS patients, undergoing mechanical ventilation, showed activation of mTORC1 in the lung epithelium via activation of the regulation of the extracellular signal-regulated kinase pathway [136]. In elderly patients with idiopathic pulmonary fibrosis, it was shown that the activation of mTORC1 signaling was elevated in fibroblasts, inducing autophagy and resistance to apoptosis, which translates into increased susceptibility to pulmonary fibrosis [137]. A study found that metabolic reprogramming induced by TGFβ treatment promoted myofibroblast differentiation and matrix production in human lung fibroblasts, in a manner dependent on mTORC1 activity; thus, mTORC1 activity may participate in exacerbated matrix protein production, contributing to the development of fibrotic lung diseases [138]. In peripheral blood mononuclear cells from COPD patients, activation of mTORC1 was found to induce corticosteroid resistance [139].

In an LPS-induced acute lung injury rat model, Emodin (a major active component of *R. officinale* from traditional Chinese medicine) improved interstitial pneumonia, interstitial microvascular edema, and inflammatory cell infiltration in lung tissues, along with the reduction of TNF-α, IL-1β, and IL-6 proteins by modulating mTOR activity [140]. A mouse model of ventilator-induced lung injury showed increased levels of lung injury markers such as IL-6 and vascular endothelial growth factor (VEGF), along with increased activation of mTORC1 in lung tissues [136]. This study also found that mice with Tsc2 gene deletion (inhibitor of mTORC1) had an exacerbation of lung injury, characterized by increased pulmonary stiffness, decreased inspiratory capacity, and impaired oxygenation; thus, activation of mTORC1 during mechanical ventilation may exacerbate lung injury [136]. In addition, mTOR has been implicated in the inflammatory response in lung injury. A study demonstrated that activation of mTOR in the lung epithelium is essential for LPS-induced inflammatory responses in a human bronchial epithelial cell line [141]. *Taraxacum mongolicum* extract reduced the inflammatory response in the lung by regulating the PI3K/Akt/mTOR axis activity in LPS-induced acute lung injury in mice [142]. Likewise, glycyrrhizic acid was shown to inhibit the production of inflammatory factors in LPS-induced acute lung injury in mice by regulating autophagy related to the PI3K/Akt/mTOR pathway [143]. Metformin decreases the inflammatory response by restoring AMP-activated protein kinase (AMPK)-dependent suppression of mTOR in endotoxemia-induced inflammatory lung injury [144].

Several studies have reported that fructose consumption can modulate mTORC1 activity in several tissues [145]. A recent study found that fructose ingestion increased mTORC1 phosphorylation and the downstream path markers of mTORC1 activity more than dietary glucose in liver tissue [146]. Moreover, fructose intake upregulates S6 phosphorylation, a marker of the mTORC1 signaling pathway, in muscle and liver tissues [147]. Another study showed that mTORC1 protein levels were increased in the pancreatic tissue of rats with diabetes induced by fructose-streptozotocin [148]. A recent report showed that fructose enhanced the production of pro-inflammatory cytokines in response to LPS by increasing mTORC1 activity in monocytes [149]. The evidence indicates that fructose can modulate mTORC1 activity in several tissues. Only one study, on idiopathic pulmonary fibrosis, has shown that fructose can indirectly promote fibrosis through the stimulation of latent TGFβ1 in human epithelial cells [150]. However, there are no reports about the effect of fructose in lung tissue (Figure 2b). Therefore, studies are needed, to elucidate the effect of fructose on mTORC1 activity in the lung and in lung disease.

## 10. Chronic High Intake of Fructose and Its Potential Involvement with COVID-19 Severity

Leading health agencies have suggested that a healthy diet improves immune system function, which could reduce risk factors for coronavirus disease 2019 (COVID-19) [151,152]. The WHO recommends limiting the consumption of added sugar to less than 10% of the total energy intake for the proper functioning of the immune system [117,118]. However, fructose consumption from SSBs increased during the COVID-19 lockdown. A cross-sectional study of 3916 US participants reported that 21.6% of adults drank more SSBs, while 9.6% drank often or always during the COVID-19 lockdown [153]. An observational study enrolling 1000 Spaniards with a mean age of 51 years reported that consumption of SSBs had increased in participants who gained weight during the COVID-19 lockdown, compared with those who did not gain weight (71.0% vs. 23.1%, *p* < 0.001) [154]; however, a COVID-19 health survey of 28,029 Belgian participants aged ≥ 18 years showed that consumption of SSBs increased both in participants who gained weight (9.2%) and in those who did not (8.7%) during the COVID-19 lockdown [155]. Data, obtained from Peruvian participants in the Perusano (pre-COVID-19) and the Stamina (COVID-19) surveys, showed a high prevalence of SSBs consumption in both populations (>78.0%) [156].

Fructose consumption from SSBs has been associated with hospitalization and mortality from COVID-19. A case-control study of 93 active-duty US Air Force members hospitalized for COVID-19 (median age of 26 years, and BMI of 25.9) found that those who consumed 3 to 6 servings/week of SSBs were more likely to be hospitalized (OR = 1.34; 95% CI: 0.61–2.92), while those who consumed ≥4 servings/day were even more likely to be hospitalized (OR = 5.23; 95% CI: 0.67–40.9) [157]. An ecological study including 158 countries found that the crude mortality rate of COVID-19 was raised with increasing consumption of SSBs (Beta: 0.340; *p* < 0.001); interestingly, it decreased by increasing fruit consumption (Beta: −0.226; *p* = 0.047), and beans and legumes (Beta: −0.176; *p* = 0.046) [158]. This indicates that high fructose consumption may exacerbate the disease severity, although molecular mechanisms associated with fructose-induced lung injury in COVID-19 patients are unknown.

There are clues about the molecular mechanisms that could be involved in the lung injury induced by severe acute respiratory syndrome coronavirus 2 (SARS-CoV-2) infection, which may be deregulated by fructose. For example, ACE2, a component of the RAS, acts as the target receptor for SARS-CoV-2, allowing virus entry and replication in host cells, including alveolar epithelial cells [159]. SARS-CoV-2 can downregulate ACE2, and the loss of pulmonary ACE2 activity is associated with acute lung injury [160]. A study including 893 COVID-19 patients reported that those chronically receiving ACE inhibitors and Ang II receptor blockers (ARB) had a lower risk of admission to the intensive care unit (OR: 0.49; 95% CI: 0.32–0.73), and death (OR: 0.23; 95% CI: 0.14–0.38), compared to those receiving no treatment [161]. In another study, COVID-19 patients receiving RAS inhibitors (hazard regression (HR) = 0.499, 95% CI: 0.325–0.767) or ARB (HR = 0.410, 95% CI: 0.240–0.700) showed a lower risk of all-cause mortality [162]. In the case of UA, a retrospective study of 1149 COVID-19 patients found that UA levels were elevated in patients who ultimately died (>400 μmol/L), compared to recovered patients (≤400 μmol/L) (OR: 3.17, 95% CI: 2.13–4.70; *p* < 0.001). UA levels were positively correlated with inflammatory markers such as ferritin, TNF-α, and IL-6 [163]. Another study reported that elevated serum levels of UA (≥423 μm/L) were associated with an increased risk of intensive care unit admission (OR: 1.8; 95% CI: 0.83–3.98), mechanical ventilation requirement (OR: 2.41; 95% CI: 1.06–5.46), and death (OR: 3.01; 95% CI: 1.16–7.81) [164]. Regarding AR activity, an open-label clinical trial with 10 COVID-19 patients found that treatment with AT-001 (AR inhibitor) was associated with reduced hospital stay and reduced mortality [165]. Moreover, in a case of SARS-CoV-2 infection in a 57-year old male, it was reported that the patient recovered after the use of AT-001 drug [166]. For the RAGE, an observational cohort study reported that COVID-19 patients had high sRAGE levels in correlation with SOFA score [*r_s_* (162) =  −0.525, *p* <  0.001] [167]. sRAGE levels were higher in patients who ultimately died [U(Ndeath  =  11, Nsurvival  =  153)  =  1520.50, z  =  4.46, *p*  <  0.001] [167]. A cross-sectional study of 145 COVID-19 patients showed elevated serum sRAGE levels in patients with severe disease (1.47 [0.97–2.25] ng/mL) compared to those with non-severe COVID-19 (0.78 [0.63–1.00] ng/mL); an association between serum sRAGE levels and COVID-19 severity was found (*r* = 0.598; *p* < 0.001) [168]. Finally, a study in 23 asymptomatic COVID-19 patients and 35 COVID-19 patients with lung involvement found that the asymptomatic patients had elevated sRAGE levels (17.5 ng/mL) compared to patients with lung involvement (2.05 ng/mL) [169].

## 11. Conclusions

In conclusion, high consumption of SSBs is related to lung damage. From emerging studies using SSB as sources of fructose, it has been shown that high intake of fructose can alter lung tissue and increase the risk of lung injuries. Studies suggest that fructose regulates the molecular mechanisms involved in lung injury. High fructose induces many low-grade inflammatory mechanisms that may contribute to lung diseases. Firstly, excessive fructose induces cellular metabolic stress that increases UA levels. UA could be an effector mechanism that triggers inflammatory response through inflammatory pathways, such as the RAS system, AR activity, and mTORC1. Secondly, chronic exposure to fructose modifies proteins (Fru-AGEs) that sustain the inflammatory state. Finally, a deleterious feedback loop sustains and may explain chronic endothelial and epithelial dysfunction, and chronic low-grade inflammation related to lung damage and dysfunction (Figure 3). Whether SSBs consumption plus infection of SARS-CoV-2 induced worsening of symptoms, the relationship of fructose consumption and other respiratory diseases must be elucidated.

## Figures and Tables

**Figure 1 nutrients-14-04089-f001:**
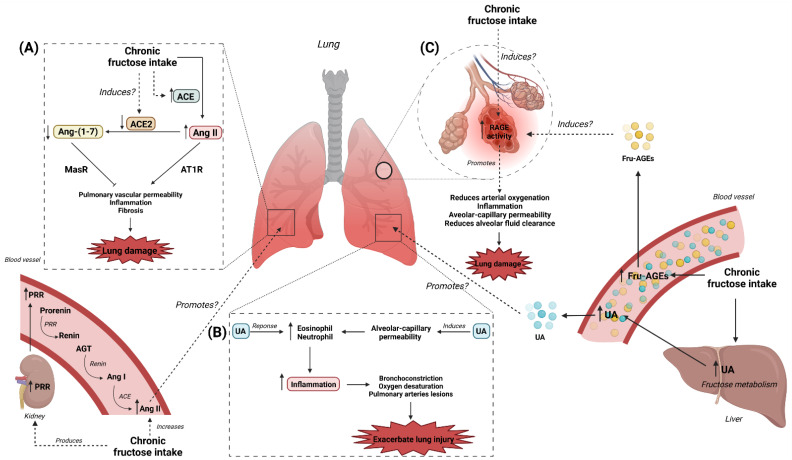
The possible role of chronic intake of fructose on lung injury through regulation of the RAS, UA, and AGEs production: (**A**) The lung damage is induced by the disbalance of the ACE/Ang II/ATR1 and ACE2/Ang-(1-7)/MasR axes [77,78]. Chronic intake of fructose induces the activation of the RAS in several tissues, including the lung. Fructose increases protein ACE and PRR [85], promoting the elevation of Ang II, while a decrease in ACE2 and Ang-(1-7) levels may induce the development of, or exacerbation of, lung injury [91,93]; (**B**) High UA levels are associated with exacerbating lung damage, promoting immune cell infiltration and inflammation [43,44,45,46]. UA is a byproduct of fructose metabolism, which is produced by purine degradation. Chronic intake of fructose increases UA levels by its metabolism in the liver [40,42], which may be a source of UA that plays an important role in the worsening of lung damage; (**C**) AGEs activate the RAGE in lung tissue, promoting lung damage by the activation of inflammation [109,110,111]. Chronic intake of fructose induces the formation of Fru-AGEs in plasma and/or tissue, which may activate the RAGE in the lung, participating in the development of, or exacerbating, the lung damage [115,116]. AGT: Angiotensinogen; ACE: angiotensin-converting enzyme; Ang II: angiotensin II; ATR1: angiotensin II receptor type 1; MasR: Mas receptor; PRR: (pro)renin receptor; UA: uric acid; AGEs: advanced glycation end products; Fru-AGEs: fructose-derived AGEs; RAGE: Receptor for AGEs. The Figure has gotten the copyright permission. Created with BioRender.com (accessed on 25 August 2022).

**Figure 2 nutrients-14-04089-f002:**
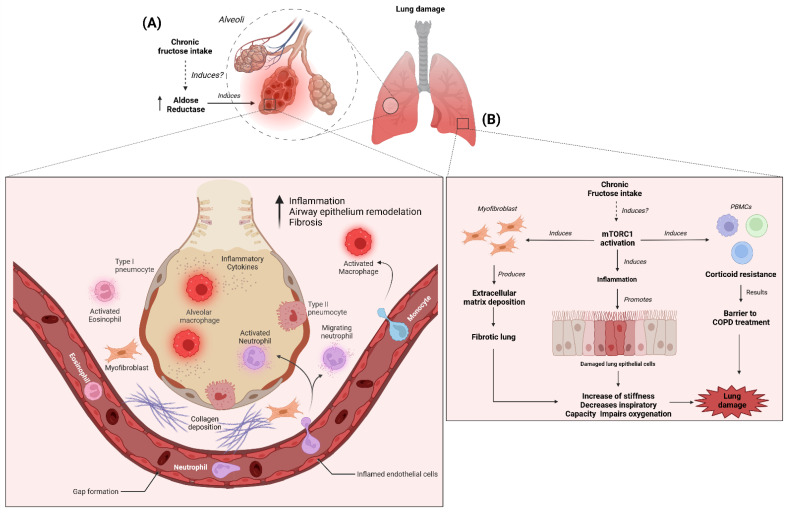
The possible role of chronic intake of fructose on lung injury through regulation of AR and mTORC1: (**A**) AR is an enzyme of the polyol pathway, that is involved in lung damage by promoting immune cell infiltration, inflammation, and fibrosis [124,125,126]. Fructose induces AR expression in several tissues [127,128,129]. Chronic intake of fructose may regulate the activity of AR in the lung, inducing lung damage through the activation of the inflammation; (**B**) mTORC1 is involved in the development of lung damage; mTORC1 induces extracellular matrix deposition and inflammation in lung epithelial cells [136,137,138]; moreover, in PBMCs, mTORC1 induces corticoid resistance, which is a barrier to lung disease treatment [139]. Chronic intake of fructose activates mTORC1 in various tissues; thus, fructose may play a role in the development or exacerbation of lung damage. AR: aldose reductase; mTORC1: mammalian target of rapamycin complex 1. The Figure has gotten the copyright permission. Created with BioRender.com (accessed on 25 August 2022).

**Figure 3 nutrients-14-04089-f003:**
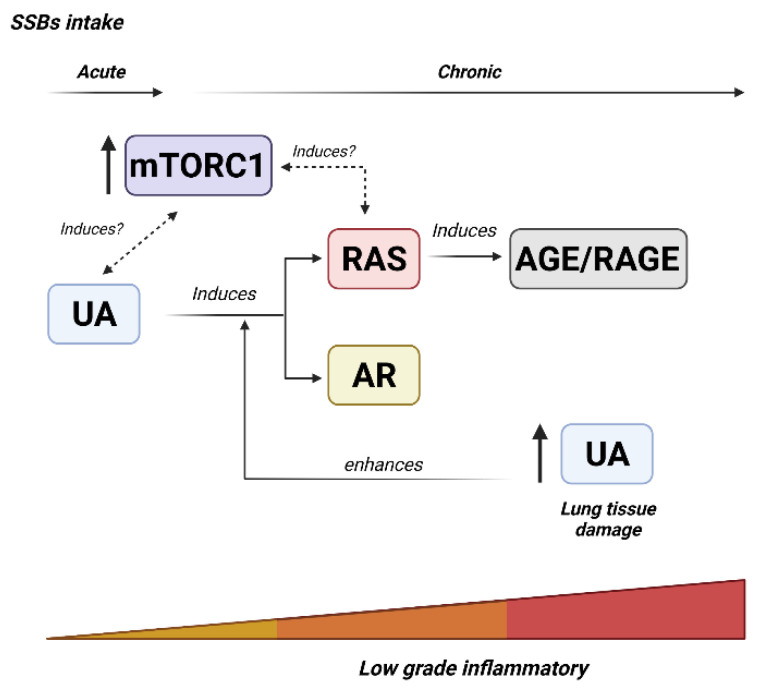
Proposed low-grade inflammatory mechanisms related to lung damage induced by SSBs intake. High intake of fructose induces many low-grade inflammatory mechanisms that may contribute to lung diseases. Firstly, excessive fructose induces cellular metabolic stress that increases UA levels. UA could be an effector mechanism that triggers inflammatory response through inflammatory pathways, such as the RAS system, AR activity, and mTORC1. Secondly, chronic exposure to fructose modifies proteins (Fru-AGEs) that sustain the inflammatory state. Finally, a deleterious feedback loop sustains and may explain chronic endothelial and epithelial dysfunction, and chronic low-grade inflammation related to lung damage and dysfunction.

## Data Availability

Not applicable.

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
