# Peer review of "Negative Effects of Chronic High Intake of Fructose on Lung Diseases"

_nutrients, 2022, doi:10.3390/nu14194089_

Round 1
Reviewer 1 Report
1. It is recommended to supplement the physiological function or metabolic pathway of fructose under normal diet, as well as the exact dose-time standard of high fructose intake and chronic fructose intake. The abstract focused on high fructose intake, whereas the text also involved in chronic fructose intake.
2: According to this article, does fructose intake present no positive physiological function at all? Could the consumption of fructose drinks also increase the risk of various diseases? Since this article repeatedly emphasizes that high or chronic fructose intake showed negative effects on the body, especially lung function, could it be interpreted that moderate or low fructose intake exhibit no side effects and may even be beneficial?
3, Whether the drawings listed in the article are original, if not, the corresponding references should be cited;
4: This article does not elaborate on the mechanism of high fructose intake on lung injury. Personal understanding: when fructose intake is too high for the body to handle, so the body would phosphorylate the excess fructose, and phosphorylated fructose has targeted negative effects on the lungs through certain receptors? If this inference is reasonable, have any studies been conducted to chemically phosphorylate fructose in vitro and then to investigate its adverse effects and mechanisms on the lungs in vivo?
5. Most of this paper is a summary of the existing literature, and there is a lack of discussion on the content of the article by the author, which is suggested to be further improved.
6: The contents in these paragraphs of this paper are scattered. Is it possible that they are related to each other and can be reflected in the conclusion?
7: The English expressions of the article can be further improved.
Author Response
Response to Reviewer 1 Comments
Point 1: It is recommended to supplement the physiological function or metabolic pathway of fructose under normal diet, as well as the exact dose-time standard of high fructose intake and chronic fructose intake. The abstract focused on high fructose intake, whereas the text also involved in chronic fructose intake.
Response 1: The title and redaction were modified to highlight chronic and high fructose intake, such as sugar-sweetened beverages. (Page 2., Line 68-103).
Point 2: According to this article, does fructose intake present no positive physiological function at all? Could the consumption of fructose drinks also increase the risk of various diseases? Since this article repeatedly emphasizes that high or chronic fructose intake showed negative effects on the body, especially lung function, could it be interpreted that moderate or low fructose intake exhibit no side effects and may even be beneficial?
Response 2: According to your questions, the beneficial functions of fructose are observed according to the fructose sources in the diet. Reports have shown that natural fructose sources such as fruit, 100% fruit juice, and others presented a protective effect on developing metabolic alterations. In contrast, SSBs as fructose source have harmful effects on metabolism, either low or high intake. Moreover, SSBs intake has been considered a significant risk factor for developing several metabolic diseases. Therefore, it was added a new section “The effect of the amount of fructose intake through SSBs on non-communicable diseases“ (Page 2., Line 67-102).
Point 3: Whether the drawings listed in the article are original, if not, the corresponding references should be cited.
Response 3: The figures listed in the manuscript were cited.
Point 4: This article does not elaborate on the mechanism of high fructose intake on lung injury.
Response 4: According to the commentary, an inflammatory mechanism related to lung damage induced by high fructose intake was added in the conclusion section. (Page 13., Line 621-628.)
Point 4: Personal understanding: when fructose intake is too high for the body to handle, so the body would phosphorylate the excess fructose, and phosphorylated fructose has targeted negative effects on the lungs through certain receptors? If this inference is reasonable, have any studies been conducted to chemically phosphorylate fructose in vitro and then to investigate its adverse effects and mechanisms on the lungs in vivo?
Response 4: According to the approach, we agree with the effect of phosphorylated fructose on lung dysfunction. No evidence indicates that phosphorylated fructose negatively affects the lung. In low and short time fructose exposure can have beneficial physiological effects against oxidative stress by producing phosphorylated derivates of fructose [1]. Current studies found that fructose-1,6-bisphosphate treatment has beneficial effects on lung damage in vitro and in vivo [2,3], however, the normal physiological functions of fructose metabolism in the lungs are still lacking. Due to the aim of this review is the effect of high-chronic fructose intake through SSBs on lung damage, the effects of phosphorylated fructose on lung tissue are not included in this review.
- Semchyshyn, H.M. Fructation in vivo: Detrimental and protective effects of fructose. Biomed Res. Int. 2013, 2013, doi:10.1155/2013/343914.
- Dias, H.B.; De Oliveira, J.R.; Donadio, M.V.F.; Kimura, S. Fructose-1,6-bisphosphate prevents pulmonary fibrosis by regulating extracellular matrix deposition and inducing phenotype reversal of lung myofibroblasts. PLoS One 2019, 14, doi:10.1371/JOURNAL.PONE.0222202.
- Jost, R.T.; Dias, H.B.; Krause, G.C.; de Souza, R.G.; de Souza, T.R.; Nuñez, N.K.; dos Santos, F.G.; Haute, G.V.; da Silva Melo, D.A.; Pitrez, P.M.; et al. Fructose-1,6-Bisphosphate Prevents Bleomycin-Induced Pulmonary Fibrosis in Mice and Inhibits the Proliferation of Lung Fibroblasts. Inflammation 2018, 41, 1987–2001, doi:10.1007/S10753-018-0842-3/FIGURES/8.
Point 5: Most of this paper is a summary of the existing literature, and there is a lack of discussion on the content of the article by the author, which is suggested to be further improved.
Response 5:: According to your suggestions, it was improved the discussion of the content in all sections of the manuscript.
Point 6: The contents in these paragraphs of this paper are scattered. Is it possible that they are related to each other and can be reflected in the conclusion?
Response 6: According to the commentary, an inflammatory mechanism related to lung damage induced by high fructose intake was added in the conclusion section. (Page 14., Line 621-628.)
Point 7: The English expressions of the article can be further improved.
Response 7: The English language was revised by us.
Reviewer 2 Report
The review article presents a novel and relevant area of study. Fructose is consumed in significant amounts worldwide and especially in the United States. The proposed hypothesis that fructose is associated with a number of lung diseases is supported throughout the paper. The proposed mechanism involving Renin Angiotensin Aldosterone system, uric acid, advanced glycation end products, aldose reductase activity, and mTORC1 activation are supported by the literature. Overall well done!
A couple minor points should be addressed.
1. A list of abbreviations would be helpful.
2. On page 2, lines 62-64, there appears to be five cities with six respective percentages of COPD. It is not clear which cities correspond to the respective percentages of COPD.
3. p 3 line 108 should be mouse model.
4. p3 line 130-131, JG cells are activated by "decreased" sodium load.
5. p4 line 182 should be "A" rather than "An."
6. Figure 1 lower left region, from the figure it appears that renin is converted into Ang 1, however, it should depict that Angiotensinogen is converted into Angiotensin 1 with renin as the enzyme. This is not clear from the picture.
Otherwise, looks good!
Author Response
Response to Reviewer 2 Comments
Point 1: A list of abbreviations would be helpful.
Response 1: According to suggestions, it was added it at the end of the manuscript (P. 15).
Point 2: On page 2, lines 62-64, there appears to be five cities with six respective percentages of COPD. It is not clear which cities correspond to the respective percentages of COPD.
Response 2: According to your commentaries, the sentence was rewritten, and added the missing city. (P. 3., L. 109-111)
Point 3: p 3 line 108 should be mouse model.
Response 3: According to your suggestion, it was changed mice by mouse. (P. 4., L. 157)
Point 4: p3 line 130-131, JG cells are activated by "decreased" sodium load.
Response 4: According to your suggestion, it was changed decreasing by decreased. (P. 6., L. 270)
Point 5: p4 line 182 should be "A" rather than "An."
Response 5: According to your suggestion, it was an by a. (P. 7., L. 331)
Point 6: Figure 1 lower left region, from the figure it appears that renin is converted into Ang 1, however, it should depict that Angiotensinogen is converted into Angiotensin 1 with renin as the enzyme. This is not clear from the picture.
Response 6: Figure 1 was revised according to your observations.
Round 2
Reviewer 1 Report
Thanks for the detailed modifications, and I have no further comments.